# Open-World Planning via Lifted Regression with LLM-based Affordances for Embodied Agents

## Abstract

Open-world planning is crucial for embodied AI agents that must make decisions with incomplete task-relevant knowledge. In fact, the main challenges lie in reasoning about objects and their affordances that are unknown to the agent. Large Language Models (LLMs), pre-trained on vast internet-scale data, have emerged as potential solutions for open-world planning. However, LLMs have limitations in long-horizon planning tasks and face problems related to interpretability, reliability, and cost-efficiency. Symbolic planning methods, on the other hand, offer structured and verifiable approaches to long-horizon tasks, but often struggle to generate feasible plans in an open-world setting. In this work, we propose a novel approach, called LLM-Regress, which combines the strengths of *lifted symbolic regression planning* with *LLM-based affordances*. The lifted representation allows us to generate plans capable of handling arbitrary unknown objects, while regression planning is the only planning paradigm that guarantees *complete* solutions using lifted representations. For such tasks, we leverage LLMs to supplement missing affordances knowledge for unknown objects. The regression nature of our approach enables the agent to focus on actions and objects relevant to the goal, thus avoiding the need for costly LLM calls for every decision. We evaluate our approach on the ALFWorld dataset and introduce a new ALFWorld-Afford dataset with higher planning complexity and more affordances types. The empirical results demonstrate that our method outperforms existing approaches in terms of success rates, planning duration, and number of LLM Tokens. Finally, we show that our approach is resilient to domain shifts in affordances and generalizes effectively to unseen tasks. This work underscores the importance of integrating symbolic reasoning with LLM knowledge for open-world decision-making in embodied AI.

## 1 Introduction

One of the biggest hurdles facing embodied agents is how to plan effectively in open-world environments with incomplete knowledge. From an object-centric perspective, making decisions in open-world environments requires reasoning about unobserved task-relevant objects and relationships. For instance, when tasked with "cleaning the room", an agent must infer the possible presence of certain objects (e.g., are there dirty plates to be cleaned?), their relationships with other objects (e.g., is the plate on the dining table or the kitchen counter?), and determine appropriate action affordances (e.g., should the plate be cleaned using a sink rather than a broom?). Given that a common household may contain thousands of items, it is impractical to predefine all relational information and action affordances for each object. Therefore, the ability to plan and make decisions with incomplete domain information is essential to developing a practical embodied agent that can be deployed in real-world settings.

Current Large Language Models (LLMs) have demonstrated promising natural language reasoning capabilities Yao et al. (2024); Ouyang et al. (2022). Many have proposed to leverage the LLMs' reasoning capabilities for planning for AI agents (Yao et al., 2022; Shinn et al., 2024). LLMs do not require structured inputs or explicit knowledge modeling, which technically makes them well-suited for open-world planning. However, growing evidence cast doubts on LLMs' capabilities for

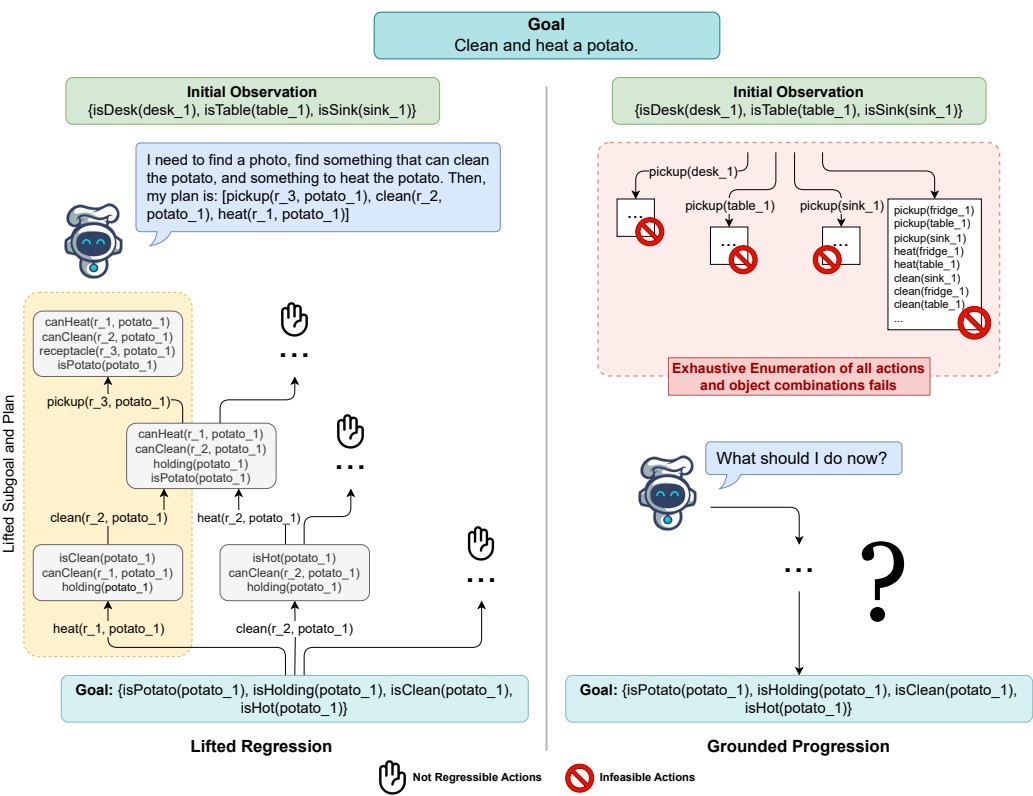

Figure 1: Lifted Regression vs Grounded Progression

long-horizon planning tasks. Additionally, LLMs are prone to hallucinations and are sensitive to prompt input Huang et al. (2023), making LLM-generated plans infeasible in practical applications that require reliability and verifiability.

Classical planning methods offer sound and complete solutions for long-horizon planning tasks. However, they require complete problem descriptions, which are not readily available in open-world scenarios. To address this limitation, many recent works have focused on leveraging the common-sense knowledge of LLMs in conjunction with symbolic planners. Notably, methods such as Silver et al. (2024); Zhang et al. (2024) use LLMs to generate closed-world planning problems (such as PDDL files) that can be solved using closed-world planners. These LLM-generated planning files can be refined through self-reflection Renze & Guven (2024) or human feedback Madaan et al. (2022) for subsequent re-planning. While these methods have achieved impressive results in some embodied AI benchmarks, they often heavily rely on similar example tasks and extensive prompt engineering. It yet remains unclear whether existing approaches can be adapted to open-world scenarios and unseen tasks. Additionally, as the number of objects, predicates, and the planning horizon increase, the problem corpus generated by LLMs can become increasingly large, resulting in the same verifiability and reliability issues for most generative models.

Most existing works on the integration of symbolic reasoning with LLM adopt the classical planning paradigm based on grounded forward search. However, these methods are designed for closed-world problems and require exhaustive enumeration of actions-object combinations. Thus, adapting classical close-world planning formalism is difficult for open-world planning which needs reasoning about unknown objects and relationships. In this work, we propose the use of Lifted Regression Planning to address open-world problems for embodied agents. Lifted representation enables us to derive plans at a structural level using variables instead of grounded objects to represent unknown objects. As shown in Fig. 1, regression planning focuses solely on relevant actions that contribute directly to the goal. This significantly reduces the search space and can produce feasible actions (a policy) for all possible scenarios. Besides, lifted regression is *complete*, meaning we can guarantee the existence of a plan (or the lack thereof), a property not assured in lifted forward searchLiu &

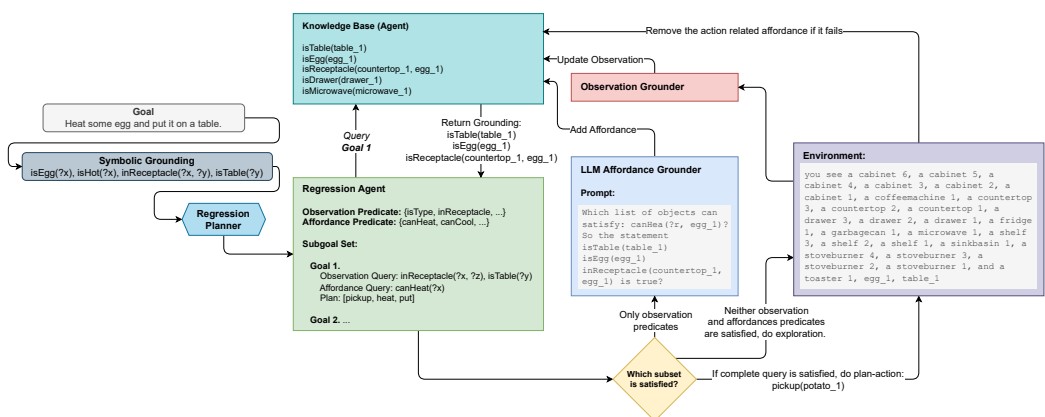

Figure 2: Lifted Regression with LLM Affordances Overview

Lakemeyer (2010). We also propose to use LLMs to generate action affordances which traditionally require manual modeling. We demonstrate that by leveraging the capabilities of LLMs, we can derive object affordances on-demand based on the lifted plan and current observations. Rather than relying on LLM to make decisions with long-horizon dependencies, our method queries LLM about a single object at a time, greatly reducing the likelihood of LLM hallucinations in a traceable manner.

We conduct experiments on the ALFWorld dataset Shridhar et al. (2020) and show significant improvement against existing baselines in terms of success rate, number of LLM query tokens and total planning time. Our methods achieved 95% accuracy on the ALFWorld while using only a small fraction of LLM tokens. To test the generalizablity of our agent, we curate the ALFWorld-Afford dataset to include more complex goals with multiple object affordances. Our empirical results show that our method is robust against shifts in affordances and can be adaptive in new tasks effectively. Overall the contributions of this work are:

- We are the first to adapt lifted regression planning to address open-world problems faced by embodied agents. Our approach provides complete and verifiable plans, guaranteeing the discovery of a feasible plan if one exists.

- We integrate LLMs into the regression planning process to derive object affordances on-demand which eliminates the need for manual efforts and offers a more robust alternative to the current LLM-based approaches.

- We introduce an extension of the existing ALFWorld dataset, called ALFWorld-Afford, which increases both the complexity of goals and object affordances to better evaluate generalizability of open-world embodied AI agents.

- We demonstrate our approach significantly outperforms the existing baselines against ALFWorld and ALFWorld-Afford in terms of success-rate and planning costs.

## 2 METHODOLOGY

Lifted regression enables plan generation for a given task involving unknown variables. The key idea is to use variables to represent unknown objects (lifting), allowing the reasoning process to proceed from a goal backward (regression). In contrast, classical planning approaches often use forward progression by enumerating all actions involving observed objects and searching forward until a goal is reached. The difference between lifted regression and grounded forward search is illustrated in Fig. 1. The benefit of lifted regression is that it does not require all relevant objects to be observed in order to generate a plan. This contrasts with grounded forward search, where no valid plan can be deduced if no actions can directly lead to the goal. In this section, we will first formally introduce lifted regression. Subsequently, we will present our own implementation for embodied agents and the integration of LLM-based affordances reasoning.

## 2.1 First Order Planning Formalism

We use the STRIPS style of definition to define our lifted regression planning problem Fikes & Nilsson (1971). We define a tuple $\Pi =< \mathcal{L}, \mathcal{A}, G, I >$ where $\mathcal{L}$ is the function-free first-order language, $\mathcal{A}$ is the set of *action schema*, $G$ is the set of *goal atoms*, and $I$ is the set of *initial state atoms*. The first-order logic language $\mathcal{L}$ contains a set of *variables* $\mathcal{V}$, a set of *constants* $\mathcal{C}$, and a set of *predicate symbols* $\mathcal{P}$. An *atom* is denoted as $p(\mathbf{t})$ where $p \in \mathcal{P}$ is a predicate symbol a vector of terms $\mathbf{t}$. The set of variables in $t$ is denoted as $vars(p)$. An atom $p(\mathbf{t})$ is a *grounded atom* if the predicate does not contain any variable, $vars(p) = \emptyset$. The name of the terms is denoted $name(t)$. A sub-goal $s = \{p_1, p_2, ..., p_i\}$ is a set of atoms that needs to be satisfied. Positive literals are a subset of $s$ including all positive literals, $s^+ = \{p \in s | p \text{ is a positive literal}\}$. Similarly, negative literals $s^-$ contains all the negative literals in $s$, i.e., $s^- = \{p \in s | p \text{ is a negative literal}\}$. A literal in $s$ must belong in either $s^+$ or $s^-$, $s = s^+ \cup s^-$. $\mathcal{A} = \{A_1, ..., A_m\}$ is a finite set of action schema. Each action schema contains three sets of atoms $A = \{pre(A), add(A), del(A)\}$. $pre(A)$ is the precondition set of atoms that must be true for the action schema $A$ to be applicable. An *action* is grounded when an action schema does not have variables in the three sets, $vars(A) = \emptyset$. In this paper, we assume that the set of action schema $\mathcal{A}$ is given and accurate. The goal $G = \{p_1, p_2, ..., p_k\}$ is given in the form of atoms for the embodied agent to satisfy.

## 2.2 Lifted Regression Algorithm

In this section, we outline the lifted regression algorithm. The algorithm is adapted from Ghallab et al. (2004) with adjustment for open-world embodied planning. First, we define the necessary operators for lifted regression as follows:

- **Substitution**: A SUBSTITUTION function $\theta(p)$ substitutes all the variables in an atom with constants or other variables. It is formally defined as $\theta(p) : \mathcal{V} \rightarrow T$ such that $\theta(p) = \{v_i \mapsto t_i | v_i \in vars(p), 1 \leq i \leq k\}$.

- **Unification:** UNIFY$(p, q)$ operator checks if two sets of predicates with different variables are equivalent Russell & Norvig (2002). It return a substitution $\theta$ if it two predicates can be unified, UNIFY$(p, q) = \theta$ where $\theta(p) = \theta(q)$. The substitution function $\theta$ is a most general unifier (MGU) whose existence indicates two predicates are equivalent. Russell & Norvig (2002) provides a detailed explanation of unification.

- **Standardization**: We use STANDARDIZE$(p)$ operator to replace all variables in $p$ with variables $v' \notin \mathcal{V}$ such that $\forall v \in vars(p), v \notin vars(\text{STANDARDIZE}(p))$. Standardization is introduced to avoid confusion between variable names for the same action schema in a plan.

- **Relevency**: RELEVANT$(A, s) = (s \cap (add(A) \cup del(A)) \neq \emptyset) \wedge (s^+ \cap del(A) = \emptyset) \wedge (s^- \cap add(A) = \emptyset)$ determines if taking an action $A$ leads to the state $s$. $A$ is relevant if the action's effects set adds something to a sub-goal without contradiction.

- **Regression**: The regression function is defined as $\gamma^{-1}(s, A) = (s - add(A)) \cup pre(A)$, it returns the previous state of $s$ before taking the action $A$. We can iterate from the goal state $n$ times to get $s_n = \gamma^{-1}(\gamma^{-1}(\theta_n(G), \theta_n(A_n)), \theta_{n-1}(A_{n-1}))...)$, where $s_n \cap I \neq \emptyset$ unless $I = \emptyset$.

The lifted regression algorithm takes in the first order language $\mathcal{L}$, finite action set $\mathcal{A}$, and the problem's goal $G$. We assume that the planning domain is acyclic and the agent initially does not observe anything thus $I = \emptyset$. The objective is to find a set $S = \{(G_1, \pi_1)....(G_n, \pi_n)\}$ which are ordered pairs of sub-goals and their corresponding plans. We also assume that no actions have been taken at the beginning of each task. For example, all the foods are not heated and all the TVs are thought to be turned off. These assumptions are made to align with the implementation of ALFWorld and most other embodied agent benchmarks. The lifted regression algorithm conducts an exhaustive backward search with each action schema $A$ until it finds all sub-goals and plan pairs when subgoals can be no longer regressed. We keep track of visited sub-goals that can be unified to prune redundant branches. The regression algorithm is shown in Algorithm 1.

To better illustrate a single-step regression, we will use a subgoal, action pair from Fig. 1 as an example. Assume that the agent's current subgoal is to heat a potato, which can be represented as a

set of predicates $G$ = {holding(potato_1), isHot(potato_1)}. A heating action is defined as heat(r, x) with $pre(a)$ = {holding(x), canheat(r, x)}, $add(a)$ = {ishot(x)} and $del(a)$ = {}. We can check that $add(a) \cap G$ = {isHot(potato_1)}, with no conflicting delete effect. Thus, we can regress $heat(x)$ with $G$ to get a new subgoal $G'$ = {canHeat(r, x), holding(x)} via $\gamma^{-1}(G, A) = (G - add(A)) \cup pre(A)$. The new subgoal indicates we want to find some object that can heat an egg.

## 2.3 LLMs as Affordances Grounding Function

For a given lifted planning problem, we can obtain a regressed sub-goal $G_n = \{p_1, p_2...\}$ which is a set of lifted predicates. When the embodied agent observes new objects, we need to check whether there are objects and a grounding that can satisfy $G_n$. We assume the agent can ground either via observations or use LLM for grounding. Thus, we define the grounding function as a composition of two functions: $\theta = \theta_o \circ \theta_a$, where $\theta_o$ is based on the agent's observation and $\theta_a$ is the grounding function from other knowledge (objects' affordances in our case). In this paper, we assume the agent has complete knowledge of $\theta_o$ to ground any observations $O = \{o_1, o_2, ..., o_k\}$, and the affordances of the objects are unknown to the agent. Rather than have predefined affordances, we want to extract them from LLM. This allows to define $\theta_a$ = LLM(PROMPT$_A$, $\theta_o(G_n)$, $O$). The function takes a predefined PROMPT, a partially grounded sub-goal $\theta_o(G_n)$, and a list of observed objects $O$ as inputs. Together they form the complete grounding needed for a regression plan.

To continue our example based on Fig. 1, where we regressed action heat(x) with $G$ = {holding(potato_1), isHot(potato_1)} to obtain subgoal $G'$ = {canHeat(r, x), holding(potato_1)}. The agent knows it is holding a potato, but does not know which objects can make canHeat(r, x) true. Assuming the agent's current observed objects are {microwave_1, kettle_1, fridge_1, counter-top_1}, we can query LLM ground predicate canHeat(r, potato_1). Assuming, that LLM returns the answer "microwave_1", we can then execute action heat(microwave_1, potato_1).

---

**Algorithm 1**
Lifted-Regression $\Sigma = \langle \mathcal{L}, \mathcal{A}, G \rangle$

1: $S \leftarrow \{\}$
2: Frontier $\leftarrow \{(G, \pi = [])\}$
3: Visited $\leftarrow \{G\}$
4: **while** Frontier is not empty **do**
5:     $G_{\text{current}}, \pi \leftarrow$ **POP**(Frontier)
6:     RegressibleActions = {}
7:     **for each** $A$ in $\mathcal{A}$ **do**
8:         $A' \leftarrow$ **STANDARDIZE**$(A)$
9:         $\theta \leftarrow$ **UNIFY**$(A', G_{\text{current}})$
10:        **if** RELEVANT$(\theta(A'), \theta(G_{\text{current}}))$ **then**
11:           RegressibleActions.add$(A')$
12:           $\pi$.append$(A')$
13:           $G' \leftarrow \gamma^{-1}(\theta(G_{\text{current}}), \theta(A'))$
14:           **if** $G'$ not in Visited **then**
15:              Visited.add$(G')$
16:              Frontier.add$((G', \pi))$
17:           **end if**
18:        **end if**
19:     **end for**
20:     If RegressibleActions = $\emptyset$ then $S$.add$((G_{\text{current}}, \pi))$
21: **end while**
22: **return** $S$

---

**Algorithm 2** LLM-Regress Agent

**Require:** $g, \theta_o, \theta_a, O, KB$
1: regress_plans $\leftarrow$ REGRESS$(g, A)$
2: FailedAff = {}
3: KB.add$(\theta_o(O))$
4: **while** $g$ is not satisfied in KB **do**
5:     **for all** $(G, \pi) \in$ regress_plans **do**
6:         **if** $\theta_o(G)$ ! = $\emptyset$ **then**
7:           aff$_G$ = LLM(PROMPT$_A$ + FailedAff, $\theta_o(s_n)$, $O$)
8:           KB.add(aff$_G$)
9:         **end if**
10:     **end for**
11:     **if** $G$ is satisfied **then**
12:         **for** a in $\pi$ **do**
13:           **if** Act(a) fails **then**
14:              FailedAff.$add$(aff$_a$)
15:           **else**
16:              progress$(G, a)$
17:           **end if**
18:         **end for**
19:     **else**
20:         $O$ = explore()
21:     **end if**
22: **end while**

---

## 2.4 Lifted Regression with LLM Affordances

Our proposed framework utilizes lifted regression to generate a set of plans and sub-goals. As shown in Fig. 2, we use a knowledge base (KB) to store facts that we observe or assume to be true. In order to distinguish predicates that are observable and the ones that need affordances reasoning, we define two sets of predicates: observation predicates $\mathcal{P}_o = \{p_{o1}, p_{o2}, \dots\}$ and affordances predicates $\mathcal{P}_a = \{p_{a1}, p_{a2}, \dots\}$. We assume that $\mathcal{P}_o$ can be grounded via observation alone. For example,

hold($eggs\_1$) is a case we assume the agent knows via feedback. $\mathcal{P}_a$ cannot be grounded by observation alone, but it is still required to check an action is feasible for some sub-goal. As previously mentioned, we rely on LLM for affordances grounding using $\theta_a$ as outlined in the previous section. If no subgoal can be satisfied, the agent randomly explores the environment. The separation between observation and affordances predicates enables us to track exactly which affordances reasoning is incorrect when an action fails. We remove affordances related to failed action from the agent's knowledge base and use it as a negative example for subsequent LLM querying. We assume that the initial observation is empty, $I = \emptyset$. The details of the algorithm are shown in Fig. 2 and Algorithm 2.

## 3 EXPERIMENTS

Our experiments are motivated by three research questions as follows:

- **RQ1:** How well does our lifted regression planning approach compare to both LLM-based planners and grounded forward search methods (with LLM-generated affordances) in terms of success rate, execution time, and the number of LLM query tokens?
- **RQ2:** How are the performance of our method and the baselines impacted against ALFWorld-Afford which has more complex goals and diverse object affordances?
- **RQ3:** Does the use of a structured knowledge base enable transferring knolwege from one task to another?

### 3.1 ASSUMPTIONS AND DESIGN DETAILS

We assume a static environment with deterministic actions. In benchmarks like ALFWorld, no actions are assumed to have occurred before the agent's execution. As a result, we assume none of the heating, cleaning, and cooling has been done on the goal object. The agent is provided with a set of action schemas, as is common in most embodied AI and robotics settings, which are detailed in Appendix C. We use a random exploration strategy, keeping track of visited and unvisited locations. Since ALFWorld goals are based on fixed templates, and existing research shows that LLMs can accurately convert these fixed template goals to PDDL goals Song et al. (2023). Therefore, we use a simple script to translate them into regression goals for this work. We use `GPT-4o` as the underlying LLM, and conducted all experiments on a computer with a modern `Intel i7` processor and `32 GB` of RAM.

### 3.2 DATASETS

We evaluate our proposed method and baseline methods on the ALFWorld Shridhar et al. (2020) and the ALFWorld-Afford benchmarks. Recall that, we curated the latter one where we increased both planning complexity and affordances types.

#### 3.2.1 ALFWORLD AND ALFWORLD-AFFORD

**ALFWorld.** ALFWorld is a text-based virtual household environment with six distinct task types: `heating`, `cleaning`, `cooling`, `pick and place`, `picking two objects and placing them`, and `examining an object under light`. The environment is partially observable where the agent need to explore to discover new items. We do not provide the agent with a predefined set of objects available in the scene; instead, objects are discovered during task execution. Initially, the agent is given access to potential locations within each room where new objects can be found. The main affordances reasoning in this environment include determining whether an object can be heated, cooled, cleaned, or turned on. Actions are deterministic, and the agent receives feedback in the form of "nothing happens" when actions have no effect. The agent has a budget of 50 actions to complete the task, and the problem is considered as a failure case if the task cannot be complete within the step limit.

**ALFWorld-Afford.** The goal of the ALFWorld-Afford dataset is to increase the planning complexity and enhance affordances reasoning diversity. While the original ALFWorld dataset provides a well-designed partially observable and open-world environment, its affordances reasoning is overly

simplistic compared to real-world scenarios. Actions like heating, cooling, and cleaning can only be performed using a microwave, fridge, and sink, respectively, which reduces the need for commonsense reasoning and instead encourages memorization of past examples. To address these limitations, we propose four additional tasks that incorporate multiple actions and affordances reasoning for various objects, making the planning domains significantly more challenging. The tasks in ALFWorld-Afford require the agents at least two object affordances with an elaborated affordances list. Detailed descriptions of these tasks can be found in Appendix B.

### 3.3 EVALUATION METRICS

We evaluate our methods against the benchmark in terms of Success Rate (SR), which is the percentage of successfully completed tasks within 50 steps. LLM-based methods like ReAct require prompting with examples and past experiences for each new action. In contrast, our method only queries for affordances when necessary. We also measure the number of *LLM tokens* for each method to assess each method's efficiency in retrieving useful information from LLMs, if one considers the potential cost of LLM calls. Additionally, we measure the average *task completion duration* to see whether each can complete a given task within a reasonable horizon. All metrics are averaged over three runs with std reproted on accuracy.

### 3.4 BASELINES

We compared our methods against the state-of-the-art LLM planners and a standard grounded PDDL planner with LLM generated affordances.

#### 3.4.1 LLM BASED PLANNERS

We employ REACT as a baseline to represent SOTA LLM planner for comparison. We implemented REACT using the original code provided by the authors which include two examples of the same task. For consistency, we use GPT-4o as the underlying LLM across all tested methods. Additionally, we explored the direct translation of action models by providing REACT with STRIPS syntax and a natural language description of our action model, assessing the LLM's ability to directly utilize a symbolic model. The prompt used for this purpose can be found in Appendix A.In scenarios involving multiple trials, we adopted Reflection to evaluate information reuse. In comparison, we allow our agent to keep track of a structured knowlege base with facts from past episode. We also investigate the ability of LLMs to generate plans without the support of REACT-style prompting. This approach is referred to as the "Standard LLM" method, where the agent is provided only with a set of instructions and plausible actions. While DEPS Wang et al. (2023a) recorded similar results as ReAct, the setup is different as action string options are provided to DEPS. The reported results for DEPS is similar to ReAct, thus we choose to only ReAct for a baseline.

#### 3.4.2 GROUNDED FORWARD PLANNER

We also designed a grounded forward planning approach using LLM-generated affordances for comparison. Instead of generating the complete domain file, we removed all affordance facts from the PDDL domain file provided by ALFWORLD and used an LLM to generate these affordance facts. Our goal is to evaluate whether generating affordances on-demand, as in our approach, offers advantages or short-comings compared to generating affordances for all objects in the domain upfront. We use a Fast Downward planner Helmert (2006) to check generate plans which is then translate to actions in the ALFWorld domain.

### 3.5 KNOWLEDGE REUSE

There is a growing interest in using feedback from previous experiences to enhance LLM-based embodied agent tasks. One notable recent work is Reflexion Shinn et al. (2024), which has shown impressive results for reflective reasoning in LLM-based agents. Our approach provides the agent with a structured memory of facts, rather than storing entire past trajectories. We specifically focus on object affordances generated by LLMs in this work. By tracking both successful and failed affordance facts, we use these examples to guide LLMs more effectively in generating affordances for new objects. We divided our experiments into two parts: (1) Similar to the Reflexion setup, we

Table 1: Performance comparison against the ALFWorld and ALFWorld-Afford benchmarks

| Method | ALFWorld | | | ALFWorld-Afford | | |
|---|---|---|---|---|---|---|
| | Success Rate | Tokens | Duration | Success Rate | Tokens | Duration |
| LLM-Regress (Ours) | **0.95±0.02** | 50K | **5 sec** | **0.84±0.03** | 62K | **8 sec** |
| ReAct w/ Examples | 0.70±0.05 | 4000K | 33 sec | 0.57±0.02 | 5600K | 41 sec |
| ReAct w/ Model Description | 0.33±0.02 | 3500K | 34 sec | 0.17±0.05 | 4200K | 39 sec |
| Standard LLM (GPT-4o) | 0.21±0.12 | 1000K | 20 sec | 0.12±0.09 | 1500K | 23 sec |
| Grounded Planner w/ LLM Afford. | 0.35±0.09 | **8K** | 13 sec | 0.29±0.07 | **8K** | 19 sec |

tracked affordances to improve the LLM's ability to reason about the same tasks, with the same goals and objects across multiple trials. (2) We also devised experiments where the agent maintained this affordance information throughout the entire ALFWorld and ALFWorld-Afford runs, allowing us to test knowledge reuse for different objects and types of tasks. This setup aims to enhance the agent's ability to generalize affordance knowledge across various tasks, resulting in improved adaptability and problem-solving efficiency.

## 4 RESULTS AND DISCUSSION

### 4.1 COMPARISON OF PLANNING APPROACHES FOR ALFWORLD (RQ1)

As shown in Table 1, our lifted regression planning approach (LLM-Regress) outperforms other baseline methods on both datasets in terms of success rate, token usage, and duration. On the ALFWorld dataset, LLM-Regress achieves a success rate of 95%, significantly outperforming other baselines. The results we obtained for ReAct are also higher (70%) than the reported results in the original manuscript, likely due to the use of GPT-4o. Regarding token usage, unlike other LLM baselines that require prompting at each step of action in addition to the potentially large base prompt, our method only requires prompting for affordances information, resulting in significantly fewer tokens used. This can lead to savings of millions of tokens, making LLM usage less costly as the number of calls adds up. This is further reflected in the planning duration for LLM-based methods, where our approach shows improved efficiency.

Another observation is that ReAct-based prompting methods heavily rely on examples. When only provided with the action model description, ReAct's performance drops drastically, indicating its inability to reason effectively without examples. Grounded planning methods are also prone to LLM hallucinations, which can lead to syntax errors and infeasible affordances. This is expected since fully specified LLM problem files contain around $4,000$ words, which increases the likelihood of such issues. These results indicate that our lifted regression planning approach is more effective and efficient than both LLM-based planners and grounded forward search methods using LLM-generated affordances. By generating affordances on demand and focusing on relevant actions, LLM-Regress reduces the computational overhead associated with exhaustive action enumeration and extensive LLM queries.

### 4.2 COMPARISON OF PLANNING APPROACHES FOR ALFWORLD-AFFORD (RQ2)

On the more complex ALFWorld-Afford dataset, which requires reasoning about a more diverse set of object affordances, LLM-Regress maintains a high success rate of 84%, while the performance of baseline methods decreases significantly. ReAct with Examples drops to a 57% success rate, and the Grounded Planner achieves only 29%. We also observed that while LLM-generated affordances might not always function correctly within the simulator, the commonsense reasoning behind them is valid. For example, the LLM might suggest using a "Counter Top" to cool a potato, which is plausible in the real world but not supported by the simulator's environment. These insights are easily obtainable with our method because we can trace LLM-generated affordances within the knowledge base, allowing for human evaluation and verification. This is in contrast to LLM prompting methods, which become difficult to interpret and manage as problems scale in complexity.

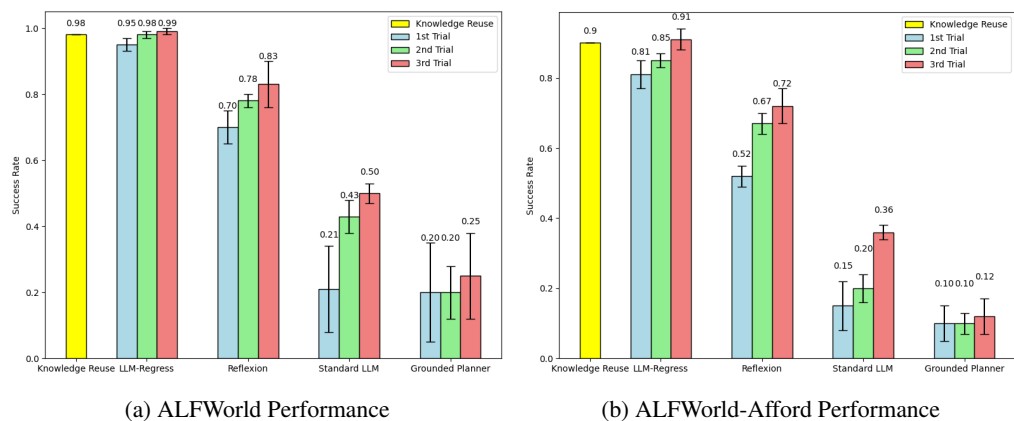

(a) ALFWorld Performance  (b) ALFWorld-Afford Performance

Figure 3: Performance comparison across different trials for various methods on the ALFWorld and ALFWorld-Afford benchmarks, including knowledge reuse metrics

## 4.3 STRUCTURED KNOWLEDGE BASE FOR KNOWLEDGE REUSE (RQ3)

Our approach leverages a structured knowledge base that allows for the easy transfer and reuse of knowledge (e.g., grounded predicates) in new tasks. This transferability enables the agent to apply previously acquired information across different contexts, including multiple trials of the same task, similar to the Reflexion setup Shinn et al. (2024). We evaluated knowledge transfer in both scenarios, and our agent showed significantly better performance as more affordances knowledge was transferred, achieving a 99% success rate on the ALFWorld dataset and 91% on the ALFWorld-Afford dataset after 3 trials of the same task. Although other LLM-based methods also demonstrated improved performance, it is difficult to determine the specific types of knowledge being reused, such as affordances information versus room layout details. In contrast, our structured approach provides clear tracking of knowledge reuse. We also evaluated the maintenance of a universal knowledge base for affordances while the agent solved the entire ALFWorld and ALFWorld-Afford datasets. This approach involved providing both positive and negative examples to guide LLMs in generating affordances candidates. We do see performance improvements ($+2\%$ on ALFWorld and $+9\%$ on ALFWorld-Afford) on both datasets when compared to treating each tasks with no knowledge transfer. Our results indicate the structured knowledge base do allow knowledge transfer in an efficient and tractable manner. Since we transfer affordances grounded in objects and types, we can easily determine whether new objects can use the same kind of affordances reasoning from the previous tasks.

## 4.4 ABLATION

In this section, we present an ablation study to evaluate the impact of affordance knowledge and the choice of LLM on the agent's performance. The *Perfect-Affordances* method provides the agent with ground truth affordances information, demonstrating the upper bound of agent performance. The results also show that affordances generated by GPT-4o are better than those generated by GPT-3.5-Turbo, indicating a better language model does improve affordances reasoning capabilities, although the impact is still less significant compared to purely LLM-based methods that require the model to perform long-horizon planning. In the *No-Affordances* scenario, all actions are made applicable to all objects, leading to a drastic drop in performance. This highlights the critical importance of affordance reasoning, as without this knowledge, the agent struggles to generate any reasonable plans and select appropriate actions to achieve the goal.

## 5 RELATED WORK

There are many recent works that investigate the planning capabilities of LLMs for decision making. Methods such as those by Wei et al. (2022); Yao et al. (2022); Renze & Guven (2024) rely on LLMs to reason about past steps using explicit reasoning prompts. Other works Ahn et al. (2022);

Table 2: Performance comparison of the impact of the amount of Affordances knowledge and different LLM choices on our method

| Method | ALFWorld | ALFWorld-Afford |
|---|---|---|
| Perfect-Affordances | 1.00 | 0.98 |
| LLM-Affordances (GPT-4o) | 0.96 | 0.81 |
| LLM-Affordances (GPT-3.5-Turbo) | 0.91 | 0.73 |
| No-Affordances | 0.12 | 0.05 |

Valmeekam et al. (2023) filter plans based on the actions or skills available to the executor, leveraging access to the internal log probabilities of the LLM. Singh et al. (2023b) proposed a structured LLM prompt framework for offline symbolic plan generation, prompting the LLM with program-like specifications of available actions and objects in the environment while using assertion checks to determine the usability of the plans.

While classical planning guarantees completeness and consistency in its solutions, it requires detailed domain descriptions, which may be unrealistic in real-world settings. Classical planning tools like PDDL (McDermott, 2000) have spurred the creation of a wide range of planning algorithms, including the Fast-Forward planner (Hoffmann & Nebel, 2001) and the BFS(f) planner (Lipovetzky et al., 2014). As a result, many recent works have attempted to combine classical planning by generating domain-specific models Arora & Kambhampati (2023); Guan et al. (2023); Xie et al. (2023); Hazra et al. (2024). These works primarily focus on generating planning domains using LLMs that can be solved by planners in the close-world setting. Our work also different from existing work that focus on learning abstract model from past experiences Konidaris et al. (2018); James et al. (2022); Ugur & Piater (2015); Ahmetoglu et al. (2022); Asai & Fukunaga (2018); Chitnis et al. (2022); Silver et al. (2023); Shah et al. (2024), as we assume the actions (skills) are predefined, but the not actual objects and their relationships.

There are also recent works focusing on open-world reasoning Ding et al. (2023); Singh et al. (2023a). These methods use LLMs to generate closed-world solutions that "could be true", based on observation. However, these approaches rely on examples to generate possible closed-world problems to guide search, without guarantees of correctness and completeness. Other work also use retrival based methods to find previous epxeriences similar to the task at handWang et al. (2023b; 2024). The reliance on examples also makes these methods difficult to generalize to new problems. To our knowledge, we are the first to focus on a formal open-world solution that guarantees completeness and makes LLM-generated knowledge traceable and verifiable.

## 6 CONCLUSION

In this work, we proposed LLM-Regress a novel approach that combines the strengths of lifted symbolic regression planning with LLM-based affordances. The lifted representation lets us generate plans capable of handling arbitrary unknown objects, while regression planning is the only planning paradigm that guarantees *complete* solutions using lifted representations. We also benefited from LLMs to reason affordances of unknown objects. We also introduced the ALFWorld-Afford benchmark which contains more complex goals and diverse object affordances compared to ALFWorld. Our experimental evaluation showed that LLM-Regress comparably outperformed known state-of-the-art solutions against the ALFWorld and ALFWorld-Afford benchmarks.

As for future work, we will enhance our solution for multi-modal settings where the agent can rely on VLMs or LMMs to segment any object and reason about their affordances, unlike ALWorld where the text is the only modality and objects are given beforehand. Furthemore, we want to include LLM commonsense knowledge to guide exploration in a more complex environment. We also want to leverage LLMs to generate new actions for unseen objects on-demand, reducing the burden of domain designers.

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

## A  EXAMPLE OUTPUT

Here is an example of output of our method in ALFWorld for a successful task

```
   -= Welcome to TextWorld, ALFRED! =-

You are in the middle of a room. Looking quickly around you, you see a cabinet
6, a cabinet 5, a cabinet 4, a cabinet 3, a cabinet 2, a cabinet 1, a
coffeemachine 1, a countertop 3, a countertop 2, a countertop 1, a drawer 3, a
drawer 2, a drawer 1, a fridge 1, a garbagecan 1, a microwave 1, a shelf 3, a
shelf 2, a shelf 1, a sinkbasin 1, a stoveburner 4, a stoveburner 3, a
stoveburner 2, a stoveburner 1, and a toaster 1.

Your task is to: heat some egg and put it in garbagecan.
go to stoveburner 2
you arrive at loc 19. on the stoveburner 2, you see a pan 1.
go to microwave 1
you arrive at loc 17. the microwave 1 is closed.
go to garbagecan 1
you arrive at loc 31. on the garbagecan 1, you see a soapbottle 2, and a tomato
3.
go to stoveburner 3
you arrive at loc 15. on the stoveburner 3, you see a pot 1.
go to shelf 2
you arrive at loc 3. on the shelf 2, you see a creditcard 3, and a saltshaker 1.
go to fridge 1
you arrive at loc 30. the fridge 1 is closed.
go to cabinet 3
you arrive at loc 20. on the cabinet 3, you see nothing.
go to stoveburner 1
you arrive at loc 15. on the stoveburner 1, you see a pot 1.
go to cabinet 4
you arrive at loc 21. on the cabinet 4, you see nothing.
go to cabinet 2
you arrive at loc 11. the cabinet 2 is closed.
open cabinet 2
you open the cabinet 2. the cabinet 2 is open. in it, you see nothing.
open microwave 1
you open the microwave 1. the microwave 1 is open. in it, you see nothing.
go to drawer 1
you arrive at loc 28. the drawer 1 is closed.
```

```
open drawer 1
you open the drawer 1. the drawer 1 is open. in it, you see a peppershaker 2.
go to countertop 3
you arrive at loc 5. on the countertop 3, you see a butterknife 3, a butterknife
 2, a butterknife 1, a cellphone 1, a cup 2, a egg 1, a fork 1, a houseplant 1,
a lettuce 3, a plate 2, a plate 1, a potato 1, a tomato 2, and a vase 3.

LLM Prompt: I am an emobided agent, I have observed a list of objects:
cabinet_6 , cabinet_5 , cabinet_4 , cabinet_3 , cabinet_2 , cabinet_1 ,
coffeemachine_1 , countertop_3 , countertop_2 , countertop_1 , drawer_3 ,
drawer_2 , drawer_1 , fridge_1 , garbagecan_1 , microwave_1 , shelf_3 , shelf_2
 , shelf_1 , sinkbasin_1 , stoveburner_4 , stoveburner_3 , stoveburner_2 ,
stoveburner_1 , toaster_1 I want to the best object or receptalce ['R']

 from my observations, so that this fact is True: inreceptacle(countertop_3,
 egg_1) & isgarbagecan(garbagecan_1) & isegg(egg_1)
 & canheat(R, egg_1)
. Please give me the best object or receptacle that would satisfy my objective.
. Please give the answer in format like best_answer:(obj_1)

LLM answer: best_answer:(microwave_1)
llm generated affordances: canheat(R, egg_1) ['microwave_1']
take egg 1 from countertop 3
you pick up the egg 1 from the countertop 3.
heat egg 1 with microwave 1
you heat the egg 1 using the microwave 1.
put egg 1 in/on garbagecan 1
you put the egg 1 in/on the garbagecan 1.
Success: True
```

Failed Affordance reasoning examples:

```
you arrive at loc 5. on the countertop 3, you see a bread 3, a
butterknife 2, a cellphone 1, a creditcard 1, a fork 2, a houseplant 1,
 a knife 2, a spatula 1, a statue 3, a tomato 3, a tomato 2, a tomato
1, and a vase 2.
LLM Prompt: I am an emobided agent, I have observed a list of objects:
cabinet_6 , cabinet_5 , cabinet_4 , cabinet_3 , cabinet_2 , cabinet_1 ,
 coffeemachine_1 , countertop_3 ,
countertop_2 , countertop_1 , drawer_3 , drawer_2 , drawer_1 , fridge_1
 , garbagecan_1 , microwave_1 , shelf_3 ,
shelf_2 , shelf_1 , sinkbasin_1 , stoveburner_4 , stoveburner_3 ,
stoveburner_2 , stoveburner_1 , toaster_1 I want to the best object or
receptalce ['R']
 from my observations, so that this fact is True: iscountertop(
 countertop_3) & inreceptacle(countertop_3, knife_2) & isknife(knife_2)
 & canclean(R, knife_2).
Please give me the best object or receptacle that would satisfy my
objective.
Please give the answer in format like best_answer:(obj_1).
LLM answer: best_answer:(countertop_3)
llm generated affordances: canclean(R, knife_2) ['countertop_3']

-----------------------------------------------------------------

LLM Prompt: I am an emobided agent, I have observed a list of objects:
glassbottle_1 , pan_2 and a list of
receptacles: cabinet_10 , cabinet_9 , cabinet_8 , cabinet_7 , cabinet_6
 , cabinet_5 , cabinet_4 , cabinet_3 ,
cabinet_2 , cabinet_1 , coffeemachine_1 , countertop_1 , diningtable_1
, drawer_2 , drawer_1 , fridge_1 ,
garbagecan_1 , microwave_1 , sinkbasin_1 , stoveburner_4 ,
stoveburner_3 , stoveburner_2 , stoveburner_1 ,
toaster_1 I want to the best object or receptalce ['R'] from my
observations, so that this fact is True:
isstoveburner(stoveburner_1) & ispan(pan_2) & inreceptacle(
stoveburner_4, pan_2) & cancool(R, pan_2).
```

```
Please give me the best object or receptacle that would satisfy my
objective.
Please give the answer in format like best_answer:(obj_1).
LLM answer: best_answer:(stoveburner_4)
llm generated affordances: cancool(R, pan_2) ['stoveburner_4']

----------------------------------------------------------------

LLM Prompt: I am an emobided agent, I have observed a list of objects:
bowl_1 , plate_1 , mug_1 , egg_1 ,
potato_1 , spatula_1 , tomato_3 , pan_1 and a list of receptacles:
cabinet_6 , cabinet_5 , cabinet_4 , cabinet_3
, cabinet_2 , cabinet_1 , coffeemachine_1 , countertop_3 , countertop_2
 , countertop_1 , drawer_3 , drawer_2 ,
drawer_1 , fridge_1 , garbagecan_1 , microwave_1 , shelf_3 , shelf_2 ,
shelf_1 , sinkbasin_1 , stoveburner_4 ,
stoveburner_3 , stoveburner_2 , stoveburner_1 , toaster_1 I want to the
 best object or receptalce ['R'] from my
observations, so that this fact is True: ispan(pan_1) & iscountertop(
countertop_3) & inreceptacle(stoveburner_2,
pan_1) & cancool(R, pan_1).
Please give me the best object or receptacle that would satisfy my
objective.
Please give the answer in format like best_answer:(obj_1).
LLM answer: best_answer:(countertop_3)
llm generated affordances: cancool(R, pan_1) ['countertop_3']
```

## B  THE ALFWORLD-AFFORD BENCHMARK

We created 150 additional tasks on top of the text version of ALFWorld by including 5 more tasks, each with at least two affordances. Additional affordances for new objects were also added to increase affordances diversity. The additional tasks and affordances are detailed below:

### NUMBER OF TASKS AND AFFORDANCE ACTIONS

| Task | Number of Tasks | Affordance Actions |
|------|-----------------|--------------------|
| pick-clean-heat-put | 30 | Heat, Clean |
| pick-clean-cook-put | 30 | Cool, Clean |
| pick-heat-cool-put | 30 | Heat, Cool |
| pick-clean-heat-put-toggle | 30 | Heat, Clean, Toggle |
| pick-clean-cool-put-toggle | 30 | Cool, Clean, Toggle |

Table 3: Number of tasks and affordance actions in the ALFWorld-Afford dataset.

### B.1  EXTRA AFFORDANCES

Additional affordances for new objects are listed below:

- **Heat:** Toaster {Bread}, Coffee Machine {Mug}, Stove Burner {Pan, Pot}
- **Cool:** Countertop {Cup, Plate, Pan, Bowl}
- **Clean:** Cloth {Apple, Egg, Cup, Pan, Tomato}, Dish Sponge {Cup, Plate, Pan, Bowl}
- **Toggle:** Microwave, Faucet, Laptop, Light Switch, Television, Cellphone, Toaster, Stove Burner

## C  ACTION MODEL

Here is the action model used in our approach defined in STRIPS actions

```
(:action PutObjectInReceptacle
  :parameters (?o - obj ?r - obj)
  :precondition (and
        ;(canContain?r ?o)
        (holds ?o)
        (holdsAny)
        (not (isContained ?o))
        )
  :effect (and
        (inReceptacle ?r ?o)
        (isContained ?o)
        (not (holds ?o))
        (not (holdsAny))
        )
)

(:action HeatObject
  :parameters (?r - obj ?o - obj)
  :precondition (and
        (canHeat ?r ?o)
        (holds ?o)
        (holdsAny)
        (not (isContained ?o))
        )
  :effect (and
        (isHot ?o)
        )
)

(:action CleanObject
  :parameters (?r - obj ?o - obj)
  :precondition (and
        (canClean ?r ?o)
        (holds ?o)
        (holdsAny)
        (not (isContained ?o))
        (not (isHot ?o))
        )
  :effect (and
        (isClean ?o)
        )
)

(:action CoolObject
  :parameters (?r - obj ?o - obj)
  :precondition (and
        (canCool ?r ?o)
        (holds ?o)
        (holdsAny)
        (not (isContained ?o))
        )
  :effect (and
        (isCool ?o)
        )
)

(:action LightObject
  :parameters (?r - obj ?o - obj)
  :precondition (and
        (canLight ?r ?o)
        (holds ?o)
        (holdsAny)
        (not (isContained ?o))
        )
  :effect (and
        (isLight ?r ?o)
        )
)

(:action ToggleObject
  :parameters (?o - obj)
  :precondition (and
        (canToggle ?o)
        (holds ?o)
```

```
        (holdsAny)
        (not (isContained ?o))
        )
   :effect (and
        (isOn ?o)
        )
)
```

## D  KNOWLEDGE REUSE RESULTS

Table 4: Performance comparison across different trials for various methods on the ALFWorld and ALFWorld-Afford benchmarks, including knowledge reuse metrics

| Method | ALFWorld | | | ALFWorld-Afford | | |
|---|---|---|---|---|---|---|
| | 1st Trial | 2nd Trial | 3rd Trial | 1st Trial | 2nd Trial | 3rd Trial |
| LLM-Regress (Ours) | **0.95±0.02** | **0.98±0.01** | **0.99±0.01** | **0.81±0.04** | **0.85±0.02** | **0.91±0.03** |
| Reflexion | 0.70±0.05 | 0.78±0.02 | 0.83±0.07 | 0.52±0.03 | 0.67±0.03 | 0.72±0.05 |
| Standard LLM (GPT-4o) | 0.21±0.13 | 0.43±0.05 | 0.50±0.03 | 0.15±0.07 | 0.20±0.04 | 0.36±0.02 |
| Grounded Planner w/ LLM Aff | 0.20±0.15 | 0.20±0.08 | 0.25±0.13 | 0.10±0.05 | 0.10±0.03 | 0.12±0.05 |
| LLM-Regress Knowledge Reuse | 0.98 | | | 0.90 | | |

