# OpenReview forum: "Open-World Planning via Lifted Regression with LLM-based Affordances for Embodied Agents"
_ICLR.cc/2025/Conference — Submitted to ICLR 2025_

### Official Review · Reviewer_qMXy · 2024-10-19

**Soundness:** 3
**Presentation:** 3
**Contribution:** 3
**Rating:** 6
**Confidence:** 2

**Summary:**

The paper addresses open-world planning for embodied AI agents facing incomplete task knowledge. It introduces LLM-Regress, which combines symbolic regression planning with LLM-based affordances to generate effective plans for unknown objects. Tested on ALFWorld and a new ALFWorld-Afford dataset, LLM-Regress outperforms existing methods in success rates, planning duration, and token usage, while also being resilient to affordance changes and generalizing well to new tasks.

**Strengths:**

1. The paper is clearly written and well-organized.
2.  The empirical evaluations conducted on both the ALFWorld and ALFWorld-Afford datasets are thorough and well-structured.
3. Symbolic reasoning and LLM are well combined, and the design is sophisticated.

**Weaknesses:**

While the LLM-Regress method is innovative, its performance is primarily evaluated within the confines of the ALFWorld datasets. More diverse environments or real-world scenarios could be explored to assess generalization capabilities. Including benchmarks against additional datasets or different planning contexts would strengthen the findings.

**Questions:**

No more questions.

---

> ### Author Response · Authors · 2024-11-23
>
> Thank you for highlighting the importance of comprehensive evaluations. While we agree that additional datasets could help assess generalization capabilities, we want to emphasize that ALFWorld and its extended dataset, ALFWorld-Afford, are particularly well-suited to our experimental setup. We need the benchmark to have a large number of objects, actions, and affordance types to effectively evaluate our approach. ALFWorld possesses several critical characteristics that make it an ideal open-world environment for embodied agent research. First, it features partial observability, where the agent operates with an egocentric view and must actively explore to discover new objects. This mirrors real-world scenarios where agents do not have complete knowledge upfront. Additionally, the environment includes 125 objects and 6 binary affordance actions, requiring reasoning over 126×2×6=93,750 potential affordances. This level of complexity aligns directly with the focus of our work, which emphasizes affordance reasoning for open-world planning. To further strengthen the benchmark, we introduced the ALFWorld-Afford dataset. This extension increases task complexity, expands the diversity of affordances, and incorporates multi-affordance tasks, thereby providing a more robust and comprehensive evaluation framework for affordance reasoning in open-world scenarios.

---

### Official Review · Reviewer_J8SY · 2024-11-04

**Soundness:** 2
**Presentation:** 2
**Contribution:** 2
**Rating:** 6
**Confidence:** 4

**Summary:**

This work proposes a lifted regression planning method that consists of an LLM-prompting step to do partial grounding while expanding the search graph. Specifically, the LLM is used for grounding affordance-related propositions. While the proposal seems interesting, the motivation and the assumptions made are not very clear, which makes it a bit hard to assess the overall contribution.

**Strengths:**

- It might be smart to use LLMs to ground the affordance information, as affordances are relatively domain-independent properties, though some care should be taken as these affordances should be compatible with the agent (e.g., SayCan).

**Weaknesses:**

- The overall weakness is that while we can more or less understand the general pipeline of the method, the motivation for doing so, the assumptions, and its limitations are not clear. Please see the below points for concrete examples.
- The comparison with the grounded planner casts a doubt on me, as I don’t understand the reason behind the difference between the two, and it’s not explained in the paper.

**Questions:**

> LLMs do not require structured inputs or explicit knowledge modeling, which theoretically makes them well-suited for open-world planning

I don’t quite get why this would make LLMs *theoretically* well-suited.


> However, they require complete problem descriptions, which are not readily available in open-world scenarios.

> While classical planning guarantees completeness and consistency in its solutions, it requires detailed domain descriptions, which may be unrealistic in real-world settings.

There are some works that focus on learning abstractions to be used for planning [1-8].


> Our approach provides complete and verifiable plans, guaranteeing the discovery of a feasible plan if one exists.

Complete with respect to the affordance performance of LLMs?

The definition of \theta_o and \theta_a—grounding functions—are vague. It’s not clear what kind of input they take and output as a result. I can only guess.

> Rather than have predefined affordances, we want to extract them from LLM.

It would be nice to motivate this argument. Also, later in the experiments:

> The main affordances reasoning in this environment include determining whether an object can be heated, cooled, cleaned, or turned on.

This seems like pre-defining affordance classes.

I believe one implicit assumption is that \theta_o should also output the labels of objects, e.g., egg_1, microwave_2, etc. Does this mean that there is a pre-defined set of object names?

We assume observation predicates \mathcal{P}_o are given but not affordance predicates \mathcal{P}_a. Why? More specifically,

> \mathcal{P}_a cannot be grounded by observation alone, but it is still required to check an action is feasible for some sub-goal.

why can we ground \mathcal{P}_o—for instance, p(hold(potato_1) | image)—but not p(canHeatStuff(microwave_1) | image)?
Relatedly, from the abstract,

> The lifted representation allows us to generate plans capable of handling arbitrary unknown objects,

If \mathcal{P}_o, which is the set of grounded propositions, is already given, then we don’t have any unknown objects?

I don’t understand how it is possible to get better results than the LLM-Affordance+FD baseline. Both the proposed method and this baseline use the affordance information generated from the LLM. So if I get this correct, the only difference in results should possibly be the planning times, since the FD should get correct plans if the operators are correct. The only way we get a difference in these results means that the domain definition should be different, which in turn means operators and/or the used propositions are different, which would further mean that the affordance-related propositions generated by the LLM are not consistent in these two approaches, LLM-regress and FD+LLM Afford.

## Some typos
- Citation typo to Russell and Norvig, AIMA, Lines 189, 192.
- Possibly typo, Line 212, “can be regressed *no* longer?”

# References
1. Konidaris, George, Leslie Pack Kaelbling, and Tomas Lozano-Perez. "From skills to symbols: Learning symbolic representations for abstract high-level planning." Journal of Artificial Intelligence Research 61 (2018): 215-289.
2. James, Steven, Benjamin Rosman, and G. D. Konidaris. "Autonomous learning of object-centric abstractions for high-level planning." Proceedings of the The Tenth International Conference on Learning Representations. 2022.
3. Ugur, Emre, and Justus Piater. "Bottom-up learning of object categories, action effects and logical rules: From continuous manipulative exploration to symbolic planning." 2015 IEEE International Conference on Robotics and Automation (ICRA). IEEE, 2015.
4. Ahmetoglu, Alper, et al. "Deepsym: Deep symbol generation and rule learning for planning from unsupervised robot interaction." Journal of Artificial Intelligence Research 75 (2022): 709-745.
5. Asai, Masataro, et al. "Classical planning in deep latent space." Journal of Artificial Intelligence Research 74 (2022): 1599-1686.
6. Chitnis, Rohan, et al. "Learning neuro-symbolic relational transition models for bilevel planning." 2022 IEEE/RSJ International Conference on Intelligent Robots and Systems (IROS). IEEE, 2022.
7. Silver, Tom, et al. "Predicate invention for bilevel planning." Proceedings of the AAAI Conference on Artificial Intelligence. Vol. 37. No. 10. 2023.
8. Shah, Naman, et al. "From Reals to Logic and Back: Inventing Symbolic Vocabularies, Actions and Models for Planning from Raw Data." arXiv preprint arXiv:2402.11871 (2024).

---

> ### Author Response · Authors · 2024-11-23
>
> We thank the reviewer for the insightful suggestions and criticisms, we want to use this opportunity to better explain some of the key concept and design choices we made in the paper.
>
> ***“LLMs do not require structured inputs or explicit knowledge modeling, which theoretically makes them well-suited for open-world planning”
> I don’t quite get why this would make LLMs theoretically well-suited.***
>
> Thank you for highlighting the inappropriate use of the word "theoretical." A more suitable term would be "technically." The rationale behind this is that LLMs process natural language as both input and output. Therefore, as long as observations and actions can be translated into natural language, LLMs are capable of reasoning about unknown objects and relationships that were not previously specified. We will update the wording accordingly in the final draft of the paper.
>
> ***”However, they require complete problem descriptions, which are not readily available in open-world scenarios.”
> While classical planning guarantees completeness and consistency in its solutions, it requires detailed domain descriptions, which may be unrealistic in real-world settings.
> There are some works that focus on learning abstractions to be used for planning [1-8].***
>
> Thank you for sharing the related works on learning abstractions for planning. We agree that these are very important contributions in model learning, and we have included a discussion of these works in the related work section. However, we aim to distinguish our approach from methods that focus on learning abstract models through past data. Our work focuses on tasks where the agent’s action model is provided but lacks information about object affordances. This mirrors robotic tasks where a robot is equipped with a set of predefined skills but does not have complete knowledge about the environment or the objects it contains.
>
> ***”Our approach provides complete and verifiable plans, guaranteeing the discovery of a feasible plan if one exists.”
> Complete with respect to the affordance performance of LLMs?***
>
> While LLMs may bottleneck our model's performance in terms of affordance reasoning, the completeness we refer to is First-Order (FO) completeness. This means our approach can generate a complete lifted plan if one exists, a property that does not hold for FO progression-style methods [1].
>
> ***The definition of \theta_o and \theta_a—grounding functions—are vague. It’s not clear what kind of input they take and output as a result. I can only guess.***
>
> \theta_o takes a set of object name strings and grounds them according to predefined goal predicates. For example, if the agent has the goal "pick up a plate" and is given the natural language description of the environment, "you are in a kitchen and you see a microwave, plate, and potato on the table,". We can use a parser of LLM to parse a list of objects O = {microwave_1, plate_1, potato_1}. \theta_o grounded O into goal-related predicates P_o = {isHolding(?o), isPlate(?o)}. Theta_o(O) = {isPlate(plate_1)}.
>
> ***“The main affordances reasoning in this environment include determining whether an object can be heated, cooled, cleaned, or turned on.”
> This seems like pre-defining affordance classes.***
>
> The affordance types are defined under the assumption that the agent has access to a set of known action schemas. In this work, we define these actions using a PDDL-like syntax. The preconditions of these actions include affordance predicates, which determine whether an action can be applied to specific objects. This framework reflects the nature of embodied agent tasks, where the agent operates with a predefined set of high-level skills.
>
> ***I believe one implicit assumption is that \theta_o should also output the labels of objects, e.g., egg_1, microwave_2, etc. Does this mean that there is a pre-defined set of object names?***
>
> We assume that \mathcal{P}_o​ is given because under our setup, the agent is capable of representing goals in symbolic form. For example, if the goal is "Give me a cleaned bowl," the agent can represent it symbolically as isBowl(?o)∧isCleaned(?o). When the agent encounters a new object in a room, it can translate its observation into text and extract the object's name. Since we use name strings for reasoning with a large language model (LLM), there is no need for a predefined set of object types. For instance, if the agent receives a natural language description of the room such as, "There is a table, chair, fridge, and XYZ in the room," it stores the set of names {’table’,’fridge’,XYZ’ }as objects. Using the LLM, the agent can reason about action affordances such as canCool(?r,potato) by grounding these names in \mathcal{P}_o​. For example, we can use LLM to evaluate canCool(fridge,potato) or canCool(XYZ,potato).

---

> ### Author Response · Authors · 2024-11-23
>
> ***“We assume observation predicates \mathcal{P}_o are given but not affordance predicates \mathcal{P}_a.” Why? why can we ground \mathcal{P}_o—for instance, p(hold(potato_1) | image)—but not p(canHeatStuff(microwave_1) | image)?***
>
> For embodied agents that have a symbolic model, we assume they must possess the ability to convert observations to symbolic forms. For example, the agent needs to identify any object and relationship within a goal for a symbolic model to reason about. If the goal is "Give me a cleaned bowl," the agent can represent it as "isBowl(?o) and isCleaned(?o)." However, defining affordance knowledge cannot often be reasoned by observation alone.  Knowledge such as affordances can be learned via prior experiences, through a manually defined database or in our case using LLM common sense reasoning. Defining affordance knowledge manually or learning such knowledge via past experience can be challenging when the number of potential objects is large. A single binary affordance can contain N^2 possible combinations for N objects. For instance, ALFWorld features 125 distinct objects and 6 affordances predicates, which require the agent to reason over a combinatorial space of 125 ^ 2 x 6=93,750 potential affordances.
>
> ***If \mathcal{P}_o, which is the set of grounded propositions, is already given, then we don’t have any unknown objects?***
>
> \mathcal{P}_o does not contain all possible grounded propositions; instead, it represents a small subset that we assume the agent can ground through observation (as illustrated in the previous examples). When the agent encounters a new object that cannot be grounded through observation, we leverage a large language model (LLM) and plans generated via lifted regression to determine whether the object is relevant to the agent's goal.
>
> ***I don’t understand how it is possible to get better results than the LLM-Affordance+FD baseline. Both the proposed method and this baseline use the affordance information generated from the LLM. So if I get this correct, the only difference in results should possibly be the planning times, since the FD should get correct plans if the operators are correct. The only way we get a difference in these results means that the domain definition should be different, which in turn means operators and/or the used propositions are different, which would further mean that the affordance-related propositions generated by the LLM are not consistent in these two approaches, LLM-regress and FD+LLM Afford.***
>
> While both LLM-Regress and FD+LLM Afford both rely on LLMs to generate object affordances, the methods for generating these affordances differ significantly. FD+LLM Afford employs a grounded forward search that requires complete knowledge of the domain to generate feasible plans. In our experiments, the LLM was provided with a complete list of objects to generate affordances for FD+LLM Afford, which need to generate 15,624 grounded affordances for a single action. This exhaustive approach often caused the LLM to struggle to produce feasible domains, resulting in frequent incorrect affordances and syntax errors.
>
> In contrast, LLM-Regress uses regression planning to identify the affordance types only relevant to the goal. LLM based affordance reasoning happens when new objects are discovered, which significantly reduces the number of affordances that need to be generated. By narrowing the scope of affordances and objects, LLM-Regress avoids the combinatorial explosion faced by FD+LLM Afford, leading to a more focused and efficient process.
>
> This efficiency is the primary reason LLM-Regress consistently outperformed FD+LLM Afford in our experiments. A detailed comparison of both methods, along with examples, is included in the Appendix.
>
> ***references***
>
> [1] Liu, Yongmei, and Gerhard Lakemeyer. "On first-order definability and computability of progression for local-effect actions and beyond." Schloss-Dagstuhl-Leibniz Zentrum für Informatik, 2010.

---

### Official Review · Reviewer_Tie4 · 2024-11-04

**Soundness:** 2
**Presentation:** 2
**Contribution:** 2
**Rating:** 5
**Confidence:** 2

**Summary:**

This work tackles open-world planning challenges for embodied AI, focusing on reasoning about unknown objects and affordances. The proposed approach, “LLM-Regress,” combines symbolic planning with knowledge from Large Language Models (LLMs) to improve efficiency and long-horizon planning. While LLMs provide affordance knowledge, symbolic planning offers structure but struggles in open-world tasks. LLM-Regress leverages LLM affordances selectively, reducing reliance on costly LLM calls and improving planning effectiveness. Tested on ALFWorld and a new, more complex ALFWorld-Afford dataset, LLM-Regress shows better success rates, faster planning, and improved generalization to unseen tasks. This approach emphasizes the benefits of blending symbolic reasoning with LLMs for open-world AI.

**Strengths:**

1. Rigorous formal reasoning may help ensure the safety and interpretability of planning.
2. Intuitively, reasoning backward from the conclusion could potentially simplify the process.

**Weaknesses:**

I am not very familiar with research on formal reasoning. However, from the perspective of developing large language models (LLMs) as agents, I feel that there is already a significant amount of similar work in this area, and the problem addressed in this paper does not appear to be particularly novel. I don’t believe that strict formal reasoning is a requirement for completing tasks in open-world environments.

1. The paper emphasizes that this method is for “open-world planning,” but it is unclear to me how this approach relates to open-world concepts. How does the paper define “open-world”? Is the evaluation environment used, ALFWorld, truly an open-world environment? In my view, open-world scenarios often imply that the objects or predicates within the environment are unknown or theoretically infinite. How would this method function in such a scenario, and what would the complexity be?
2. What is “grounding”? The paper lacks a clear definition.
3. The assumption that “the affordances of the objects are unknown to the agent” is puzzling. If the next action or subgoal is determined primarily by prompting a large model, what is the fundamental distinction between this approach and previous works like SayCan [1]?
4. Figure 1 is not very clear. I suggest adding a more detailed caption to explain concepts such as “Lifted Regression” and “Grounded Regression” so that readers don’t have to refer to the main text for clarification.
5. The experiments are only conducted in ALFWorld, which limits the paper’s persuasiveness.
6. The DEPS [2] framework has also been tested on ALFWorld. A discussion and comparison with DEPS in relevant sections would strengthen the paper.

[1] Do as i can, not as i say: Grounding language in robotic affordances
[2] Describe, Explain, Plan and Select: Interactive Planning with LLMs Enables Open-World Multi-Task Agents

**Questions:**

Refer to weakness.

---

> ### Author Response · Authors · 2024-11-23
>
> We thank the reviewer for the insightful suggestions and criticisms, we want to use this opportunity to better explain some of the key concept and design choices we made in the paper.
>
> ***The paper emphasizes that this method is for “open-world planning,” but it is unclear to me how this approach relates to open-world concepts. How does the paper define “open-world?”***
>
> We adopt the formal logic definition of "open-world," where "the facts asserted in a model are not assumed to be complete" [3]. In our setup, this concept translates to assuming that certain elements of the model (such as object types, locations, and affordances) are incomplete or unknown. In our setup, we assume knowledge of the agent's action models in a symbolic form. This mirrors many real-world robotic tasks, where robots are typically programmed with a predefined set of skills but lack comprehensive knowledge of their environment, including the objects they interact with, their locations, and the affordances required for every task. By addressing this partial knowledge scenario, our approach aligns with the principles of open-world reasoning while tackling practical challenges in robotic planning.
>
> ***Is the evaluation environment used, ALFWorld, truly an open-world environment?***
>
> ALFWorld exhibits several characteristics that make it an open-world environment well-suited for embodied agent research. First, it incorporates partial observability: the agent operates with an egocentric view and must actively explore its surroundings to discover objects, reflecting the incomplete knowledge typical of open-world settings. Furthermore, the environment features 125 distinct objects and 6 binary affordance actions (e.g., canHeat(toaster, egg), canCool(fridge, bread)), requiring reasoning over a combinatorial space of 125×6×2=93,750 potential affordances. This complexity underscores the open-world nature of ALFWorld, as the agent must interact with objects in a partially observable environment with incomplete knowledge.
>
> ***In my view, open-world scenarios often imply that the objects or predicates within the environment are unknown or theoretically infinite. How would this method function in such a scenario, and what would the complexity be?***
>
> It is an intriguing prospect to build systems that can act in an infinitely large environment with no prior knowledge of any objects and relationships. The method we proposed in the paper is more suitable for high-level reasoning for any agent than other formal methods. The key benefit of lifted regression is that we produce plans containing objects and predicates that are relevant to a goal. Also, since the plan we produce is in a lifted form, our method only focuses on objects that satisfy certain properties (which can be reasoned about using LLMs). As a result, our method greatly reduces the number of objects and predicates the agent needs to reason about. This is in contrast to grounded forward search methods, where the agent needs to plan over all possible objects in the environment. (A benefit of our approach is highlighted with SayCan[2] in the next two questions.)
>
> ***What is “grounding”? The paper lacks a clear definition.***
>
> The grounding we refer to is the first-order logic definition of grounding: “Grounding is the task of taking a problem specification, together with an instance, and producing a variable-free first-order formula representing the solutions to the instance" [4]. In the context of ALFWorld, it is the process of determining which objects the agent has seen to make a subgoal true. For example, the agent sees a set of objects {apple, microwave, plate}, and the agent needs to determine which object can ground so the predicate canHeat(?x, ?y) to make it true. In grounded forward search methods, actions also need to be grounded first before planning. For example, the action pick(?x) with 500 objects will produce 500 grounded actions {pickup(apple), pickup(plate), ...}. We will use SayCan as an example in the next question to contrast it with our approach.

---

> > ### Author Response · Authors · 2024-11-23
> >
> > ***The assumption that “the affordances of the objects are unknown to the agent” is puzzling. If the next action or subgoal is determined primarily by prompting a large model, what is the fundamental distinction between this approach and previous works like SayCan [1]?***
> >
> > Thank you for highlighting the need for a comparison with works like SayCan, as it underscores important distinctions between our approaches, reflecting the difference between reasoning with grounded representations compared to lifted representations. First, SayCan assumes full observability of the environment, where the exact location and state of all objects are known. ALFWorld on the other hand is partially observable. SayCan relies on an LLM to reason about each grounded action and select the most suitable one. This requires pre-grounding all possible actions, which scales poorly as the number of objects increases. This limitation is evident in their experimental setup, where the agent completes tasks in environments with 15 objects, and 3 actions. This is significantly fewer than the number of objects typically found in an ALFWorld environment, which contains 125 objects and 9 actions. SayCan reasons in grounded space, meaning that at each planning level, it must reason about 551 grounded actions (as reported in their paper), compared to just 9 in our approach. If we were to adopt a grounded approach for ALFWorld, we would need to reason about at least 93,750 × 9 grounded actions for each step. The number of actions also increases exponentially with the planning horizon, making such an approach infeasible for long-horizon tasks.
> >
> > ***Figure 1 is not very clear. I suggest adding a more detailed caption to explain concepts such as “Lifted Regression” and “Grounded Regression” so that readers don’t have to refer to the main text for clarification.***
> >
> > We acknowledge the lack of clarity in Figure 1 and revised it to include a detailed caption explaining key concepts such as lifted regression and grounded regression. Additionally, we improved the figure's readability to a larger fond to ensure it is accessible to readers without requiring reference to the main text.
> >
> > ***The experiments are only conducted in ALFWorld, which limits the paper’s persuasiveness.***
> >
> > Our work focuses on open-world reasoning in partially observable environments with unknown object affordances, requiring domains with an egocentric view and a rich class of object affordances. While benchmarks like Minecraft[1] offer a rich environment with diverse object types, they are less suitable for our focus on affordance reasoning. Minecraft’s primary affordances revolve around mining and crafting actions. Mining assumes tools can mine predefined objects, which limits the need for affordance reasoning. Similarly, crafting relies on recipes that specify exact item combinations, making it more rule-based and less open-ended. This focus on predefined mechanics contrasts with ALFWorld’s requirements, where agents must reason about whether specific actions can apply to objects based on their affordances. To further enhance the benchmark, we introduced the ALFWorld-Afford dataset, which increases task complexity, expands affordance diversity, and incorporates multi-affordance tasks, providing a more robust evaluation framework for affordance reasoning.
> >
> > ***The DEPS [2] framework has also been tested on ALFWorld. A discussion and comparison with DEPS in relevant sections would strengthen the paper.***
> >
> > We thank the reviewer for highlighting these related works. We included a discussion with DEPS them in the discussion and incorporated their reported ALFWorld results for comparison in our revised manuscript. DEPS shares a similar framework with the ReAct methods used as baselines, as both adopt a think-plan-reflect loop. While DEPS emphasizes plan feasibility, ReAct focuses on plan refinement.
> >
> > ***citations***
> >
> > [1] Wang, Zihao, et al. "Describe, explain, plan and select: Interactive planning with large language models enables open-world multi-task agents." arXiv preprint arXiv:2302.01560 (2023).
> >
> > [2] Ahn, Michael, et al. "Do as i can, not as i say: Grounding language in robotic affordances." arXiv preprint arXiv:2204.01691 (2022).
> >
> > [3] Vacca, John R., ed. Computer and information security handbook. Newnes, 2012.
> >
> > [4] Aavani, Amir, et al. "Grounding formulas with complex terms." Advances in Artificial Intelligence: 24th Canadian Conference on Artificial Intelligence, Canadian AI 2011, St. John’s, Canada, May 25-27, 2011. Proceedings 24. Springer Berlin Heidelberg, 2011.

---

> > > ### Comment · Reviewer_Tie4 · 2024-11-26
> > >
> > > Thank you for your response. I will improve my original rating.

---

### Official Review · Reviewer_Ae1D · 2024-11-09

**Soundness:** 3
**Presentation:** 3
**Contribution:** 2
**Rating:** 5
**Confidence:** 4

**Summary:**

Manuscript presents an approach of combining canonical symbolic planning methods with LLMs to tackle the open-world challenge to classic planners and the inability of long-horizon planning of LLMs. The key idea is to replace all hand-crafted affordance functions with queries to LLMs, while still using the pre-defined list of objects and predicates (perception or grounding is assumed to be offered by the environments). The proposed method is evaluated against several LLM planner counterparts and its own ablations on ALFWorld and a variant.

**Strengths:**

+The manuscript is mostly clear and well-written. The research topic (open-world planning) is of interest to the NeurIPS community.

+The proposed method is technically sound. Indeed the component that requires most ontology engineering in canonical planning frameworks is the affordance functions and it makes sense to replace it with LLM queries. While I believe the perception part is also complicated and can be replaced as well, given the benchmark (ALFWorld) offers ground truth symbolic observations, I think it is ok to leave it as is.

+The empirical results look strong, as compared to the picked LLM planners. The ablation studies can be relatively simple but it delivers the points on the crucial roles of affordances.

**Weaknesses:**

My major concerns rely on the evaluation side:

-More benchmarks should be adopted. Albeit the reputation of ALFWorld in planning and LLM agents, it is not very "open-world" as the number of objects, functionalities, tasks, etc, can be relatively limited. My suggestion is to extend to some truly open-world environments, ex. Minecraft [1]. It adds more complexity as the environment is more volatile and necessitates much more background knowledge. I would like to see some results on this benchmark.

-More baselines, especially LLM planning methods should be compared. Notably, the community has already noticed that straightforward planning with LLMs does not usually work very well and several remedies have been proposed: interactive planning with self-check or reflections[1], adding memory as a means of learning[2], retrieval-augmented planning with LLMs[3], etc. I think [1] also has results on ALFWorld. A more comprehensive comparison of these approaches could help strengthen the merit of the proposed method.

-Misc: I found the font size of the graphics in the manuscript can be too small to see clearly.


[1] https://arxiv.org/abs/2302.01560

[2] https://arxiv.org/abs/2311.05997

[3] https://arxiv.org/abs/2403.05313

**Questions:**

See "Weaknesses".

---

> ### Author Response · Authors · 2024-11-23
>
> We thank the reviewer for the insightful suggestions and criticisms, we want to use this opportunity to better explain some of the key concept and design choices we made in the paper.
>
> ***More benchmarks should be adopted. Albeit the reputation of ALFWorld in planning and LLM agents, it is not very "open-world" as the number of objects, functionalities, tasks, etc, can be relatively limited.***
>
> We thank the reviewer for the insightful comment. However,  in this work, we focus on the formal logic definition of "open-world," where "the facts asserted in a model are not assumed to be complete" [1]. In our setup, we assume the object types, locations and affordance of these objects are incomplete parts of our model.  ALFWorld demonstrates several key characteristics that make it an open-world environment suitable for embodied agent research. First, it features partial observability, where the agent operates with an egocentric view and must explore to discover new objects. This setup mirrors real-world scenarios where agents lack complete knowledge upfront. More importantly, the environment includes 125 objects and 6 binary affordance actions(i.e canHeat(toaster, egg)), requiring reasoning over 125 ^ 2 \* 6 =93,750 potential affordances. This complexity aligns with the focus of our work, which centers on affordance reasoning for open-world planning. To further enhance the benchmark, we introduced the ALFWorld-Afford dataset, which increases task complexity, expands affordance diversity, and incorporates multi-affordance tasks, providing a more robust evaluation framework for affordance reasoning.
>
> ***My suggestion is to extend to some truly open-world environments, ex. Minecraft. It adds more complexity as the environment is more volatile and necessitates much more background knowledge. I would like to see some results on this benchmark.***
>
> We thank the reviewer for pointing out the comparison, we added [2] in the related work section. While Minecraft[2] offers a rich environment with diverse objects types, it is less suitable for our focus on affordance reasoning. Minecraft’s primary affordances revolve around mining and crafting actions. Mining assumes tools can mine predefined objects(that are domain specific to Minecraft), which limits the need for affordances reasoning. Similarly, crafting relies on recipes that specify exact item combinations which is also specific to the Minecraft domain. This focus on domain specific reasoning which contrasts with ALFWorld's household setup that mimic real-world senarios. We recognize Minecraft’s potential to complement our work in multi-modal multi-modal reasoning, enabling us to explore the integration of LLMs with visual and textual inputs. These extensions would allow us to evaluate our framework in richer and more complex open-world scenarios.
>
> ***More baselines, especially LLM planning methods should be compared. Notably, the community has already noticed that straightforward planning with LLMs does not usually work very well and several remedies have been proposed: interactive planning with self-check or reflections[1], adding memory as a means of learning[2], retrieval-augmented planning with LLMs[3], etc. I think [1] also has results on ALFWorld. A more comprehensive comparison of these approaches could help strengthen the merit of the proposed method***
>
> We thank the reviewer for highlighting these related works. We also included a discussion about memory-based methods, such as those in [3,4], in the related work section. We also included them in the discussion and incorporate the reported ALFWorld results for comparison in our revised manuscript. DEPS shares a similar framework with the ReAct methods used as baselines, as both adopt a think-plan-reflect loop. The use of external memory as a mechanism to learn or reason with LLMs is an intriguing and important approach for enhancing LLMs' planning capabilities. However, our method focuses on model-based reasoning, which does not rely on external memory, making it not directly comparable to memory-based methods. That said, we believe extending our work to incorporate the extraction of common-sense knowledge from past data presents an intriguing direction for future research. This could potentially bridge the gap between model-based reasoning and memory-augmented approaches, further improving planning capabilities in open-world environments.
>
>
> ***Citation***
>
> [1] Computer and information security handbook. Newnes, 2012.
>
> [2] "Describe, explain, plan and select: Interactive planning with large language models enables open-world multi-task agents." arXiv preprint arXiv:2302.01560 (2023).
>
> [3] "Jarvis-1: Open-world multi-task agents with memory-augmented multimodal language models." arXiv preprint arXiv:2311.05997 (2023).
>
> [4] "Rat: Retrieval augmented thoughts elicit context-aware reasoning in long-horizon generation." arXiv preprint arXiv:2403.05313 (2024).

---

> > ### Author Response · Authors · 2024-12-02
> >
> > Hi there, this is a friendly reminder that today is the last day for reviewer comment, and we’d love to hear your feedback or address any questions you might have.

---

### Meta-Review · Area_Chair_uWcS · 2024-12-25

**Metareview:**

This paper proposes a framework that combines lifted symbolic regression planning with LLMs to tackle open-world planning tasks. By using lifted representations, the approach can generate complete and verifiable plans for tasks involving unknown objects, while LLMs are used on-demand to infer missing affordance knowledge. This design helps minimize frequent reliance on LLM queries, making the process more cost-efficient. The authors also introduce ALFWorld-Afford (an extension of the ALFWorld dataset) featuring complex goals and a broader range of object affordances (to test generalizability). Through experiments, the paper tests LLM-Regress vs. baselines and shows it performs better in success rates, planning time, and LLM token usage.

Reviewers appreciated the novel and technically sound approach of using LLMs to generate affordances on-the-fly and integrating them into a lifted symbolic regression planner, highlighting its potential to make LLMs useful in embodied AI contexts. They noted that the framework is built on a rigorous and formal structure, which supports safety and interpretability in planning.

(+) Originality: Ae1D and particularly J8SY praised the novelty. In the post-rebuttal discussion, it was noted that "the method is built upon a formal structure."

In their reviews and post-rebuttal discussion, the following consensus emerged:

(-) Clarity: Tie4, J8SY, and the AC, after careful reads, found that the writing of the manuscript requires significant improvement. The committee found that technical terms were not clearly defined, assumptions were inadequately explained, contextualization with respect to crucial prior work was largely insufficient, and figures were difficult to interpret.

(-) Significance: While the method is original and interesting, reviewers largely agreed that more empirical substantiation would be necessary to deliver a convincing and impactful result at ICLR standards. The authors provided arguments for why certain widely-adopted open-ended / planning benchmarks and memory-based baselines were not ideally suitable for LLM-Regress. However, the committee finds that demonstrating the impact of a formal solution across more than a single standardized benchmark (and authors' proposed extension) would enhance the work's impact.

The AC observed that only minor updates were made to the manuscript, leaving substantial feedback from the ICLR discussion phase unincorporated. Additionally, reported results and token counts in Table 1 were modified by the authors without notifying the committee, though these appear to be primarily writing errors. Finally, the AC concurs with the reviewers that the manuscript requires significant improvements, particularly in its writing, technical clarity, and the introduction of concepts and methodology.

In summary, due to issues with clarity (primary) and empirical substantiation (secondary), this work requires a significant revision. Once these changes are addressed, a fresh round of reviews will be necessary for thorough evaluation.

**Additional Comments On Reviewer Discussion:**

* Ae1D: maintained 5
* Tie4: 3 -> 5
* J8SY: 3 -> 6
* qMXy: initial score of 6.


qMXy expressed low confidence in their review, submitted a very brief evaluation, and did not acknowledge the author rebuttal or participate in the private post-rebuttal discussion within the committee. The AC primarily weighed the detailed feedback and arguments from Ae1D, Tie4, and J8SY. Reviewers spent considerable effort sharing their feedback, which could potentially lead to improved clarity of text and figures, better coverage of crucial prior works, and clearer definitions of technical terms.


However, only minor updates were made to the manuscript, leaving substantial feedback from the ICLR discussion phase unincorporated. Additionally, reported results and token counts in Table 1 were modified by the authors without notifying the committee (though these appear to be primarily writing errors).


In the post-rebuttal discussion within the committee, no reviewer unequivocally championed the submission.


The typos, while still abundant in the revised version, have been given no weight in this decision-making.

---

### Decision · Program_Chairs · 2025-01-22

Reject